Candidal carriage in saliva and subgingival plaque among smokers and non-smokers with chronic periodontitis—a cross-sectional study

Santhana Krishnan Gayathri 1
Naik Dilip 1
Uppoor Ashita ashita.uppoor@manipal.edu 1
Nayak Sangeeta 1
Baliga Shrikala 2
Maddi Abhiram 3
1 Department of Periodontology, Manipal College of Dental Sciences Mangalore, Manipal Academy of Higher Education , Manipal , Karnataka , India
2 Department of Microbiology, Kasturba Medical College Mangalore, Manipal Academy of Higher Education , Manipal , Karnataka , India
3 Periodontics & Endodontics, School of Dental Medicine, State University of New York at Buffalo , Buffalo , NY , United States of America
Folayan Morenike
Electronic publication date: 2020 Jan 29
Publication date: 2020
Volume: 8
Electronic Location ID: e8441
Received 2019 Aug 23; Accepted 2019 Dec 19
Copyright: ©2020 Santhana Krishnan et al.
Copyright year: 2020
Copyright holder: Santhana Krishnan et al.
License: This is an open access article distributed under the terms of the Creative Commons Attribution License, which permits unrestricted use, distribution, reproduction and adaptation in any medium and for any purpose provided that it is properly attributed. For attribution, the original author(s), title, publication source (PeerJ) and either DOI or URL of the article must be cited.
License URL: https://creativecommons.org/licenses/by/4.0/

Keywords: Candida carriage, Periodontitis, Subgingival plaque, Saliva, Smoking

Funding: The authors received no funding for this work.

==============================
Background and Objectives

Studies of gum or periodontal disease have focused mainly on bacterial pathogens. However, information related to fungal species in the saliva and subgingival mileu is particularly lacking in smokers with periodontitis. This cross-sectional study compared the prevalence of various Candida species in saliva and subgingival plaque samples of smokers and non-smokers with periodontal disease.

Methodology

Study subjects were recruited into three group—Group 1: Smokers with chronic periodontitis (N = 30), Group 2: Non-smokers with chronic periodontitis (N = 30) and Group 3: Healthy controls (N = 30). Clinical parameters recorded included plaque index (PI), gingival index (GI), periodontal probing depth (PPD) and clinical attachment loss (CAL). Saliva and subgingival plaque samples were collected from subjects from the above groups. The collected samples were processed for isolation and identification of various Candida species using CHROMagar chromogenic media. Additionally, antifungal susceptibility tests were performed for the isolated Candida species in order to assess antifungal drug resistance to fluconazole and voriconazole.

Results

Prevalence of Candida species in saliva samples was quantified as 76.6% in Group 1, 73.3% in Group 2 and 36.6% in Group 3 and statistically significant differences were observed between groups 1 & 3. Prevalence of Candida species in subgingival plaque samples was quantified as 73.3% in Group 1, 66.6% in Group 2 and 60% in Group 3 and no statistically significant differences were observed between groups. Candida albicans was the most frequently isolated species followed by Candida krusei and Candida tropicalis. A positive correlation was observed for smoking exposure, pack years and Candida colonization. A marginally significant positive correlation was observed between Candida colonization and increasing pocket depth and attachment loss. Antifungal drug resistance was mainly observed for Candida krusei in both saliva and subgingival plaque samples.

Conclusion

Based on the results we can conclude that oral candidal carriage is significantly increased in smokers with periodontal disease. Mechanistic studies are needed to understand the importance of Candida species in periodontal disease.

Introduction

Periodontal disease is a multifactorial disease and associated with complex microbial interactions. Periodontopathogenic bacteria, mainly red-complex bacteria (Porphyromonas gingivalis, Treponema denticola, and Tannerella forsythia) and Aggregatibacter actinomycetemcomitans have been implicated as the etiological agents for periodontal disease (Socransky & Haffajee, 2005). However, recent studies that performed microbiome sequencing indicate that more than 700 species of bacteria are part of the oral milieu (Griffen et al., 2012; Abusleme et al., 2013; Oliveira et al., 2016; Liu et al., 2012). Some of these species are beneficial and contribute to a healthy oral homeostasis, while others are associated with disease. Keystone species are considered as pathogenic components of the disease-inducing microbiota (Stone & Xu, 2017). Proinflammatory characteristics and interspecies signaling result in community shifts in microbiota along with a host response leading to periodontal tissue destruction (Lopez, Hujoel & Belibasakis, 2015; Andreski & Breslau, 1993). However, a third of the oral microbial species-level taxa remain uncultured, including many of the disease-associated microbial organisms (Andreski & Breslau, 1993). This indicates that in addition to red complex bacteria other species may play a role pathogenesis of periodontitis.

There has been a rising interest in understanding the role of fungi in dental plaque and periodontal disease. Recent microbiome analyses have indicated that Candida albicans is the most abundantly found opportunistic fungus in the saliva as well as subgingival plaque (Ghannoum et al., 2010; Vesty et al., 2017). Many predisposing factors (local and general) cause an increase in Candida colonization causing damage to the oral mucosa. They include cigarette smoking, poor oral hygiene, hypovitaminosis, dentures, pregnancy, HIV infection, diabetes mellitus and debilitated patients on antibiotics, steroids or cytotoxic therapy  (Muzurović et al., 2013). In a healthy state, Candida species reside on the buccal mucosa, tongue, palate, and saliva. They have been recovered in 40–60% of healthy oral carriers. Although candida colonization is rarely found in subgingival sites in healthy subjects, Candida species have often been isolated from periodontal pockets (Samaranayake, 2009). Few earlier studies explored the occurrence and probable role of Candida species in periodontal diseases (Samaranayake, 2009; Canabarro et al., 2013; McManus et al., 2012). These studies used culture methods and/or genotypic methods to isolate and identify Candida species in the oral and plaque samples. They concluded the increased presence of C. albicans in periodontal pockets as compared to healthy sites, although there was no statistical significance. However, C. albicans was found to be associated with severity of periodontal disease (Samaranayake, 2009; Canabarro et al., 2013; McManus et al., 2012).

Smoking is a major environmental factor that significantly increases the risk of periodontal disease (Tomar & Asma, 2000; Johnson & Hill, 2004). Smoking is known to affect periodontal attachment loss via several pathways including microcirculatory, inflammatory and immune-mediated mechanisms. However, the exact mechanisms of the effect of smoking at the cellular and molecular level are not clear (Nociti Jr, Casati & Duarte, 2015). Smoking has long been considered to affect the microbial milieu by favoring periodontal pathogenic bacteria. However, recent studies have shown that there is no significant difference in the supragingival or subgingival microbial milieu between smokers and non-smokers (Nociti Jr, Casati & Duarte, 2015; Kibayashi et al., 2007). Most of these studies have focused on bacterial species and the data on fungal species is particularly lacking. Besides the relationship between smoking, periodontitis and oral Candida colonization is unclear. Moreover, microbial profiling in aggressive periodontitis showed an increased prevalence of Candida in patients who were smokers (Kamma, Nakou & Baehni, 1999). Hence this study aimed to assess the quantitative and qualitative oral colonization of Candida species in saliva and subgingival sites among smokers & non-smokers with chronic periodontitis.

Materials & Methods

Study participants

The present cross-sectional study was performed in the Department of Periodontology, Manipal College of Dental Sciences, Mangalore Manipal Academy of Higher Education, Manipal, Karnataka, India. A total of 90 male patients in the age range of 20–50 years (mean age 35.88  ± 8.39 yrs) reporting to the outpatient section were recruited based on the following criteria between Jan 2015 to Jun 2016. This was due to the extremely low incidence of smoking in females as seen in the Indian population Inclusion criteria for this study were as follows: patients with moderate to severe generalized chronic periodontitis (i.e., >5 mm loss of attachment, >4 mm probing depth in at least 30% of the sites, bleeding upon probing present in mouth) (American Academy of Periodontology, 1999) (Armitage, 2004). Smokers included those who have smoked 100 cigarettes in a lifetime while those who never smoked were included in the non-smoker group (Andreski & Breslau, 1993). The exclusion criteria were as follows: subjects who were on medication such as corticosteroids, antibiotics, or medication for xerostomia, antifungal agents/antiseptic mouthwash over past six months; subjects who reported any systemic predisposing factor for oral candidiasis such as diabetes mellitus or anemia, those wearing removable dental prosthesis or orthodontic appliance, subjects using smokeless tobacco and subjects having aggressive periodontitis, necrotizing ulcerative gingivitis or necrotizing ulcerative periodontitis. Approval was obtained from the Institutional Ethics Committee and Review board, (Protocol no: 14129). All the participants signed the informed consent, preceding, commencement of the study following the Helsinki agreement.

Patient groups

After the sample size calculation, a non-probability convenience sampling was done, the study participants were divided into three groups of 30 participants each: Group 1: Smokers with periodontitis, Group 2: Non-smokers with periodontitis, Group 3: Healthy patients (non-smokers) with no periodontal disease (controls).

Smoking status & nicotine dependence

Smoking history was elicited from the patient based on a questionnaire. Smoking exposure was calculated as the number of cigarettes per day × duration (in years). Pack years were calculated according to the formula: (No. of cigarettes smoked/day) * (No. of years smoked)/(No. of cigarettes in one pack). Fagerstrom test was performed to assess the intensity of physical dependence on nicotine (Heatherton et al., 1991). Nicotine dependence of subjects was assessed based on smoking habits and frequency (Heatherton et al., 1991).

Clinical examination

A complete clinical examination of full mouth was done using a periodontal probe (Williams’s markings; Hu-Freidy, Chicago, IL, USA) and recorded by two experienced & calibrated examiners for each patient. The following clinical parameters were recorded: Plaque index (PI)  (Silness & Loe, 1964), Gingival index (GI) (Loe & Silness, 1963), Probing pocket depth (PPD), Clinical attachment loss (CAL), Bleeding on probing (BOP)-Modified sulcus bleeding index (Mombelli et al., 1987): Probing pocket depth and loss of clinical attachment at six sites around each tooth (mesio-buccal, buccal, disto-buccal, mesio-lingual, lingual, and disto-lingual) were assessed to the nearest millimeter using Williams’s periodontal probe. Clinical attachment loss was measured in millimeters from Cemento Enamel Junction (CEJ) to the base of the periodontal pocket. All measurements were taken by a single examiner.

Microbiological sampling and analysis

Sample collection

The sample collection was done by the same clinical examiner between 9 AM to 11 AM. Subjects were refrained from food intake, drink, any oral hygiene procedure or smoking for at least 1 h before sample collection. Saliva collection was done as previously described using concentrated oral rinse (COR) technique (Nikawa et al., 1993). Briefly, each study participant was provided with 10 ml of sterile phosphate-buffered saline (PBS 0.1M, pH 7.2) and asked to rinse their mouth thoroughly for 60 s and then to expectorate in a sterile plastic tube. The collected samples were processed immediately in the Microbiology department of the Kasturba Medical College of Mangalore. Subgingival plaque was collected as previously described  (McManus et al., 2012). For each patient, after superficial cleaning of the sites with cotton pellets and removal of supragingival plaque the subgingival plaque samples were collected from the three sites with the deepest periodontal pockets by using Gracey curettes and pooled into one container. The collected subgingival plaque samples were transferred to PBS in sterile capped containers and transported on ice to the Microbiology department for microbiological analysis of Candida species.

Assessment of Candida prevalence & relative abundance

Quantification of candida carriage in the samples was performed as described previously (Darwazeh, Al-Dwairi & Al-Zwairi, 2010). Briefly, samples were centrifuged for 10 min at 1,000 rpm and the supernatant discarded. The pellet was re-suspended in 1 ml of PBS. A sample volume of 20 microlitre was taken and streaked using a sterile glass spreader onto Sabourauds Dextrose Agar plates in duplicates for culture and incubated at 37 °C. After 48 h, the Candida colonies were counted and the colony-forming unit per ml (CFU/ml) of saliva and subgingival plaque was calculated. Candida sub speciation was determined using CHROMagar cultures. The Candida isolates from the culture plates were inoculated onto CHROMagar plates and incubated at 37 °C for 48 h. Different Candida species imparted different colors to colonies when incubated in CHROMagar chromogenic medium: C. albicans—light green colonies with pale edges; C. parapsilosis—pale cream colonies; C. krusei—spreading rose pink colonies with broad, pale edges; C. tropicalis—bluish-green colonies; C. glabrata—dark pink colonies with pale edges. After determination of Candida subspecies, antifungal susceptibility testing was done for the isolated species.

Antifungal disc diffusion susceptibility testing

Each sample isolate was subcultured on Sabouraud dextrose agar and incubated at 35 °C twice before testing to ensure clarity and optimum growth. For antifungal testing the Clinical and Laboratory Standards Institute (CLSI) reference method for broth dilution antifungal susceptibility testing of yeasts was used (NCCLS, 2002). Briefly, Mueller-Hinton agar enhanced with 2% glucose and 0.5 g of Methylene Blue (MB) per ml were used and discs were applied to determine susceptibility. In distilled water, standard solutions of glucose (0.4 g/ml) and MB (5 mg/ml) were added and the medium was prepared. They were sterilized and stored at 4 °C. The Mueller Hilton agar plates were made using the Glucose Methylene Blue (GMB) standard solution (2.9 ml) and allowed to absorb for 4 to 6 h before inoculation. The inoculum was prepared using 4–5 representative colonies of the Candida sp., and suspending in 5ml of sterile normal saline to match 0.5 McFarland standard. Candida albicans ATCC 90028 strain was utilized as reference strain for disk diffusion testing. A sterile cotton swab dipped into inoculum was inoculated onto agar plates by evenly streaking in all directions over the whole surface area of the agar plates. The plates were allowed to dry for 15 min and the disks with fluconazole and voriconazole are placed onto each inoculated plate, and the plates were incubated at 35 °C. Then readings were taken after 24 and 48 h. Inhibitory zone (zone of inhibition) diameters for the disks were measured and interpreted as follows: Susceptible—Zone diameters of 19 mm seen, Susceptible-dose dependent—Zone diameters of 15 to 18 mm seen, Resistance—Zone diameters of 14 mm seen (Barry & Brown, 1996). The readings were taken after 24 and 48 h. The diameter of zone of inhibition was measured at the transitional point where growth abruptly decreases and was determined by reduction in colony size, number and density.

Statistical analysis

Data within each Group were expressed as mean  ± standard deviation (SD) in 90 patients. All calculations were performed using SPSS software v 20.0 (SPSS Inc., Chicago, IL, USA). Comparison of the clinical parameters and microbiological parameters between the control and two experimental Groups were performed by one way ANOVA Post hoc power analysis was done by applying the Tukey test. A p-value < 0.05 was considered as statistically significant. Association of smoking as a confounding factor with Candida colonization and periodontitis was performed using Pearson’s correlation test.

Results

Clinical parameters of periodontal disease

Since the study participants were age-matched, the comparison of age distribution within the Groups showed no statistically significant difference. The mean plaque index (PI) score of Groups 1, 2 and 3 was 1.62  ± 0.48, 1.51  ± 0.51 and 0.67  ± 0.22 respectively. No significant differences were seen for PI. This suggests that the effect of tobacco smoking on the periodontium were independent of the plaque level. The gingival index (GI) for Groups 1, 2 and 3 was 1.53  ± 0.29, 1.64  ± 0.43 and 0.79  ± 0.18 respectively. There was a statistically significant difference for GI between Groups 1 and 2. The bleeding on probing (BOP) score for Groups 1, 2 and 3 was 1.6, 1.9 and 1.03 respectively and was found to be significantly different between all Groups. The probing pocket depth (PPD) among Groups 1 and 2 was 6.64  ± 0.60 mm and 6.85  ± 0.85 mm respectively. The comparison of PPD within the Groups showed no significant difference between Groups 1 and 2. The clinical attachment loss (CAL) among Groups 1 and 2 was 7.06  ± 0.71 mm and 7.18  ± 1.17 mm respectively. The comparisons of CAL showed no statistically significant difference between Groups 1 and 2.

Relative abundance of Candida species in saliva and subgingival plaque samples

Candida species were isolated from 76.6% of patients in Group 1, 73.3% in Group 2 and 36.6% of patients in Group 3 from saliva samples. The mean CFU/ml was 13  ± 13.7, 10  ± 12.2 and 3.7  ± 3.1 in Groups 1, 2 and 3 respectively. When compared among the Groups, Candida colonization showed statistically significant differences among Groups 1 and 3 (p < 0.004). This data indicates that in saliva, Candida species are significantly higher in smokers with periodontitis in comparison to healthy controls. Candida species were isolated in 73.3% subjects in Group 1, 66.6% in Group 2 and 60% in Group 3 from subgingival plaque. The mean CFU/ml was 8.1  ± 7.383, 7.47  ± 8.83 and 4.43  ± 4.88 and in Groups 1, 2 and 3 respectively. However, statistical analyses did not indicate any significant differences in subgingival plaque samples between Groups. A total of 5 different Candida subspecies were isolated among the 3 study Groups from saliva and included C. albicans, C. tropicalis, C. krusei, C. glabrata and C. parapsilosis. However, only 4 different subspecies were isolated from subgingival plaque and included C. albicans, C. krusei, C. tropicalis and C. glabrata. In saliva, C. albicans was the most commonly isolated subspecies among all Groups, while C. krusei and C. tropicalis were next followed by C. glabrata which was the least frequently isolated species. Interestingly, C. parapsilosis was isolated from saliva of only one subject in the smokers with periodontitis Group (Group 1). Even in subgingival plaque, C. albicans was the most commonly isolated subspecies among all Groups. C. krusei and C. tropicalis were the second and third most frequently isolated subspecies for Groups 1, 2 and 3 respectively from subgingival plaque. Among Groups 1 and 2, C. glabrata was isolated from 3.30% subjects. (Table 1 and Fig. 1). The antifungal susceptibility testing showed that the majority of the isolated Candida subspecies were susceptible to the drugs, fluconazole and voricanozole. Interestingly, C. krusei isolates showed antifungal resistance in both saliva and subgingival plaque samples (Table 2).

Table 1 Relative abundance of various Candida species within various groups for saliva and subgingival plaque samples.

Isolated samples were processed for determining CFU/ml of various Candida species within groups.

Candida species	Saliva	Subgingival plaque	
	Group 1n (%)	Group 2n (%)	Group 3 n (%)	Group 1 n (%)	Group 2 n (%)	Group 3 n (%)	
C. albicans	15(50)	14(46.7)	13(43.3)	10(33.3)	11(36.7)	13(43.3)	
C. parapsilosis	1(3.30)	0	0	0	0	0	
C. krusei	3(10)	4(13.30)	3(10)	7(23.30)	4(13.30)	3(10)	
C. tropicalis	3(10)	4(13.3)	3(10)	4(13.30)	4(13.30)	2(6.7)	
C. glabrata	1(3.3)	0	0	1(3.30)	1(3.3)	0	
No Candida	7(23.3)	8(26.6)	11(36.6)	8(26.6)	10 (33.3)	12(40)	
Notes.

n no. samples positive for Candida species

Figure 1 Candida subspeciation in saliva samples from smokers with periodontitis.

Isolated saliva samples were centrifuged to pellet the microbial isolates which were then cultured on CHROMagar chromogenic media. The development of specific colored colonies helps in identifying the species of Candida within the sample.

Table 2 Antifungal susceptibility of Candida isolates from subgingival plaque and saliva samples of various groups.

Isolated samples were processed for determining antifungal susceptibility for fluconazole and variconazole.

Candida species	Saliva	Subgingival plaque	
	Sensitive	Resistant	Sensitive	Resistant	
	Count (n)	%	Count (n)	%	Count (n)	%	Count (n)	%	
C albicans (F)	44	97.8	1	2.2	34	100	0	0	
C albicans (V)	44	100	0	0	34	100	0	0	
C parapsilosis (F)	1	100	0	0	0	0	0	0	
C parapsilosis (V)	1	100	0	0	0	0	0	0	
C krusei (F)	4	57	3	42.9	6	35.3	11	64.7	
C krusei (V)	6	100	0	0	17	100	0	0	
C tropicalis (F)	9	100	0	0	11	100	0	0	
C tropicalis (V)	9	100	0	0	11	100	0	0	
C glabrata (F)	0	0	0	0	2	100	0	0	
C glabrata (V)	0	0	0	0	2	100	0	0	
Notes.

F Flucanazole treated

V Voricanozole treated

n number of samples

% the percentage of samples that are either sensitive or resistant to antifungal agents

Correlation among clinical parameters and microbiological analysis of samples

Correlation tests were performed to examine the correlation between smoking exposure, nicotine dependence, pack year, PPD and CAL. There was fair correlation between smoking exposure, (p 0.311) nicotine dependence, (p 0.291), pack years (p 0.150) and Candida colonization in saliva and subgingival sites. This data indicates that duration of smoking and number of cigarettes smoked correlated positively with Candida colonization. There was also positive marginally significant correlation between Candida colonization and increased PPD and CAL. Deeper pocket depths correlated with increased prevalence of complex Candida species (Table 3, Figs. 2 and 3).

Table 3 Karl Pearsons Correlation between Candida colonization in saliva and subgingival plaque and smoking.

Pearson’s correlation coefficient (r-value) was calculated for understanding the correlation between the above characteristics. Candida colonization was originally calculated based on CFU/ml for each sample.

Parameters	Candidacolonizationin Saliva (r-value)	Candidacolonization in Subgingival Plaque (r-value)	
Fagerstorm test	0.291(Fair)	0.118	0.091	0.631	
Smoking exposure	0.311(Fair)	0.094	0.311(Fair)	0.130	
Pack years	0.150	0.429	0.141	0.141	
PPD	−0.100	0.447	−0.022	0.868	
CAL	−0.57	0.663	0.038	0.773	
Notes.

PPD Probing Pocket Depth

CAL Clinical Attachment Loss

Figure 2 Correlation of Candida colonization of saliva with smoking and periodontitis.

Pearson correlation coefficient was calculated for understanding the correlation between the above characteristics.

Figure 3 Correlation of Candida colonization of subgingival plaque with smoking and periodontitis.

Pearson correlation coefficient was calculated for understanding the correlation between the above characteristics.

Discussion

In the current study, an association between the subgingival colonization of Candida species, especially C. albicans, in smokers with periodontitis was noted. In addition, a great diversity of Candida species, such as Candida tropicalis, Candida krusei, Candida parapsilosis, Candida glabrata were also associated with Candida albicans as observed in previous studies (Canabarro et al., 2013). This further confirms that advanced forms of periodontitis and tobacco smoking are associated with complex yeast communities in deep periodontal pockets. Also, a great diversity of Candida species, such as C. tropicalis, C. krusei, C. parapsilosis, C. glabrata were found to be associated with C. albicans. Past studies that assessed the microbial profile of smokers in aggressive periodontitis patients found increased levels of C. albicans in periodontal pockets  (Kamma, Nakou & Baehni, 1999; Barry & Brown, 1996). Furthermore, tobacco users were found to have elevated levels of C. albicans in saliva (Sheth et al., 2016).

In the present study, smoking was found to be independent of plaque levels in agreement with various other studies (Tanaka & Tanaka, 2010; Lie et al., 1998). Clinical signs of gingival inflammation such as bleeding, redness, and exudation were not evident in smokers with periodontitis (Group 1). This could be due to peripheral vasoconstriction of blood vessels caused by smoking. The localized vasoconstrictive actions of nicotine may also be responsible for the reduced vascular supply in gingiva. Increased sites with bleeding on probing were observed in Group 2 in agreement with previous studies (Urzúa et al., 2008; Tanaka & Tanaka, 2010; Lie et al., 1998).

Saliva harbors large concentration of Candida species in the disease process and provides a niche and reflects the changes in nature and behavior of the underlying disease process (Palmer et al., 1999). Saliva can be a good means of identifying oral candidal carriage. Besides, it has been speculated that the percentage of Candida species in the subgingival plaque is proportional to some bacterial periodontopathogens, suggesting the probable role for Candida species in the pathogenesis of periodontal disease (Thein, Samaranayake & Samaranayake, 2007). In the present study, the overall candidal carriage was significantly higher in the saliva of smokers with periodontal disease concurrent to previous studies (Muzurović et al., 2013; Darwazeh, Al-Dwairi & Al-Zwairi, 2010; Lie et al., 1998). Low salivary flow rate and increased dryness of the mouth due to long term smoking altered host response and vasculature as seen among smokers. This, in turn, may favor the increase in Candida colonization. In subgingival plaque samples, Candida colonization was seen among all the groups concurrent to earlier studies (Canabarro et al., 2013; McManus et al., 2012). However, comparison between the groups was not statistically significant. This could be attributed to the relatively lower amounts of subgingival plaque samples collected as compared to the saliva samples and also the individual variations among different study subjects. Candida density was also very heterogeneous in the saliva and subgingival plaque samples. These differences could be attributed to the altered immunological status due to smoking and chronic systemic inflammation from periodontitis in these subjects similar to earlier studies (Bakki et al., 2014; Kleinegger et al., 1996).

In the present study, C. albicans was the most commonly detected Candida species in saliva and plaque samples, which is similar to data from previous studies (Kleinegger et al., 1996; Sardi et al., 2011; Barros et al., 2008; Jewtuchowicz et al., 2008). Decreased colonization of C. albicans was seen in subgingival plaque compared to saliva. This could be due to site-specificity and minimal amount of species which go undetected. C. glabrata, C. tropicalis, C. krusei, and C. parapsilosis were the other candida subspecies found in the present study. C. krusei and C. tropicalis were higher in Group1 & 2 (Table 1) similar to earlier studies (Javed et al., 2014; Reichart et al., 2002). This suggests that this species may be particularly adapted to oral colonization as a constituent of normal human oral flora, with a potential to cause clinical infection. Studies have shown that C. krusei and C. tropicalis are more virulent, possibly due to their capacity to adhere to epithelial cells in vitro and secretion of proteinase (Järvensivu et al., 2004). It was also noteworthy that specific Candida subspecies obtained in saliva sample were not seen in subgingival plaque samples of the same subject. This shows that individual sites are colonized by distinct Candida species (site-specific) which are otherwise not seen on the oral mucosa or periodontal pockets, or in healthy patients as observed previously  (McManus et al., 2012).

The Fagerstrom test revealed high nicotine dependence (score of 8–10) in 10 patients and moderate dependence score (5–8) in 6 patients and also that Candida colonization increased with nicotine dependence. An increase in smoking duration and number of cigarettes showed an increase in Candida colonization in saliva and subgingival plaque (Table 3). Cigarette smoking possibly increases keratinization of epithelium and enhancement of hydrophobicity and is recognized as a predisposing factor for oral yeast carriage (Williams et al., 1999). There was a marginal positive correlation in candida colonization with increasing pocket depth and attachment loss. This may be due to a small sample size but is concurrent with other studies (Canabarro et al., 2013; McManus et al., 2012; De-La-Torre et al., 2018).

C. albicans has a predominant role in the immune evasion and adhesion to the periodontal tissues. The periodontal pockets provide a suitable environment for change in microflora and predispose periodontal destruction. C. albicans may exacerbate periodontitis by enhancing the invasion of host cells by anaerobic bacteria such as P. gingivalis. P gingivalis, can alter the local immune response and hypothetically may influence opportunistic organisms like C. albicans inhabiting the same subgingival niche and thus there could be a symbiotic and synergistic relationship between the two organisms. Moreover, C. albicans can produce proteinases that destroy major extracellular matrices and basement membrane components (Järvensivu et al., 2004). The current study indicates that smoking and periodontitis affects oral colonization of Candida species. Smoking impairs host responses to periodontal therapy. Additionally, a recent study in a rat model revealed that long term smoking attenuates host defense against C. albicans by suppressing NLRP3 inflammosome (Ye et al., 2019). Periodontal pathogens combined with some Candida species are resistant to short-term periodontal therapy (Williams et al., 1999). Furthermore, C. albicans has been shown to increase antibiotic tolerance of oral plaque bacteria like Streptococcus gordonii (Chinnici et al., 2019). Additionally, it was also found that binding of S. gordonii to the cell wall of C. albicans was important for biofilm formation and antibiotic tolerance of S. gordonii (Chinnici et al., 2019). Antifungal susceptibility testing, indicated that C. krusei was found to be resistant to fluconazole, in both subgingival plaque and saliva concurrent to previous studies (Tumbarello et al., 1998; Cartledge, Midgley & Gazzard, 1999). But the other Candida s ubspecies isolated in this study were not susceptible to either fluconazole or voricanozole. It needs to be examined if this antifungal susceptibility of oral Candida species is variable depending on genetic and geographical factors. Such information may be useful in predicting the occurrence of systemic candidiasis.

A major limitation of this study is related to the gender of the subjects. Only male subjects were included here as the number of women smokers is low in India with a prevalence of 4% (Rani et al., 2003). Also, since it is cross sectional study, the cause and effect relationship cannot be established. Given the findings of the current investigation, further large multicenter studies are required to examine the effect of smoking on oral candidal carriage in standardized study populations with long term follow up and interventional treatment. Also, future studies should incorporate variables such as salivary flow, saliva composition, and Candida adhesion to oral epithelial cells for a more comprehensive analysis.

Conclusion

From the analysis of the results and within limitations of the present study, it can be concluded that oral candidal burden is increased in smokers with chronic periodontitis as compared to smokers without periodontal disease. It needs to be investigated further if this increase in candidal burden in smokers with chronic periodontitis has any relationship with etiopathogenesis of periodontal disease in this population.

Supplemental Information

Supplemental Information 1 Master Chart for Group 1 Subjects

Click here for additional data file.

Supplemental Information 2 Master Chart for Group 2 Subjects

Click here for additional data file.

Supplemental Information 3 Master Chart for Group 3 Subjects

Click here for additional data file.

Additional Information and Declarations

Competing Interests

Author Contributions

Human Ethics

Data Availability

Abhiram Maddi is an Academic Editor for PeerJ.

Gayathri Santhana Krishnan, Dilip Naik and Ashita Uppoor conceived and designed the experiments, performed the experiments, analyzed the data, prepared figures and/or tables, authored or reviewed drafts of the paper, and approved the final draft.

Sangeeta Nayak analyzed the data, prepared figures and/or tables, authored or reviewed drafts of the paper, and approved the final draft.

Shrikala Baliga conceived and designed the experiments, performed the experiments, analyzed the data, authored or reviewed drafts of the paper, and approved the final draft.

Abhiram Maddi analyzed the data, authored or reviewed drafts of the paper, and approved the final draft.

The following information was supplied relating to ethical approvals (i.e., approving body and any reference numbers):

The Institutional Review Board, MCODS, Mangalore, Maniapl Academy of Higher Education granted ethical approval to carry out the study within its facilities (Protocol No 14129).

The following information was supplied regarding data availability:

Raw data is available as Supplemental File.

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
