# Peer review of "Candidal carriage in saliva and subgingival plaque among smokers and non-smokers with chronic periodontitis—a cross-sectional study"

_PeerJ, doi:10.7717/peerj.8441_

## Round 0.1 · original submission · Major Revisions

· Academic Editor

Major Revisions

Please do read carefully the reviewers comments and make notes of your response to each comment

·

Basic reporting

Please see attached PDF page with all comments to the authors.

Experimental design

The experimental work was well conducted. Though, some explanations are needed to justify the use of morphological characters only to identify Candidal isolates? and why the authors only chose to test very few antifungal agents? Other comments are highlighted in the enclosed PDF report.

Validity of the findings

Please see attached comments.

Additional comments

Please make sure to check spelling and language.

Reviewer 2 ·

Basic reporting

1.The rules in scientific naming of organisms should be adhered to
2. Rephrase the statement in line 287-291 and clearly describe a point at a time. The presence of diversity of Candida spp. should be discussed separately from the point of C. albicans dominating with clear reference of each.
3. Should only discuss the results presented. Since female were not included in the study line 291-296 should be deleted.
4. It is important to explain how did the stated limitations affect the study findings for reproducible of the study
5. With citation clear description of the relationship between smoking and C. krusei and C. tropicals is warranted
6. Conclusion should built from discussion that based on the study findings. Line 384-387 should be rephrased

Experimental design

1. More clarification on the pooled subgingival pockets samples. How were they pooled, how many subgingngival pockets were pooled together, what happened in the culture positive pooled samples
2. Any scientific explanation of excluding female participants even when the population of smokers is pronounced to be low they still had the right to participate. Wont there be any interesting finding in this small
3. More description of the inclusion criteria in line 143-145 are warranted. Did the investigator considered all the mentioned criteria for each patients? Were there any superior criteria in recruitment?
4. Methodology need more detail. Clear description of Fagerstom test in line 166-167 is required
5. With reference clear description on how were the measurement taken in line 169-177 used in the study. What was concluded if some measurement were abnormal while others are normal in a single patients? Which measurement weight more than others
6. More details on methodology line 221-224 regarding any discrepancies between the 24 hrs readings and 48 hrs reading. Which reading was considered, How many times were the test repeated in case of discrepancies
7. What was the considered level of confidence interval on statistical analysis
8. What was the considered normal range of the mean plaque index

Validity of the findings

1. For easy understandings of the reader all the reports on the abstract should be reported with frequency and percentage
2. All statistical significant data should be presented with their statistical test and strength
3. Quantitative part which is highly pronounced in the title is neglected in the abstract section
4.More details on the meaning of fair correlation and marginally significant correlations as used in line 278-280. Statistical numbers representing the terms should be used

Additional comments

1. Authors should be carefully with scientific naming of bacteria
2. In abstract the repeated world "as" should be deleted
3. It is not clear what was the sample size. Was it 90 or 30 more clarifications are needed
4. Sentence should not start with number
5. Several grammatical errors need corrections before publications example use of capital letter in the middle of sentence and small letters at the beginning of the sentences
6. some words are joined and lose the meaning example in abstract, conclusion section "....althoughC albicans ...."

---

## Round 0.2 · Major Revisions

· Academic Editor

Major Revisions

Dear Author,

Please do address the issues raised by the peer reviewer. We look forward to reviewing your final manuscript

·

Basic reporting

The article is clearly written with good background and context being provided.

Experimental design

The methodology was well defined with sufficient information. Some questions are however still need to be answered as shown in the attached comments document.

Validity of the findings

The conclusions need to be revised. The authors concluded that smoking significantly contributed to the colonization effect by Candida among their study group. However, the same trend was evident among the non-smokers. Only when you compare these two groups with the healthy group, then there seem to be some differences and it is mainly in the saliva samples.

Additional comments

Please see the attached document.

---

## Round 0.3 · accepted · Accept

· Academic Editor

Accept

Thanks for submitting your interesting manuscript to PeerJ

·

Basic reporting

The article is well written, clear and conform with professional standards.

Experimental design

The experiments were well designed and executed.

Validity of the findings

The data presented was very sound and controlled. Conclusions were linked directly to the questions raised and were supported by the findings of the results.